# The Multi-Faceted Nature of Renalase for Mitochondrial Dysfunction Improvement in Cardiac Disease

**DOI:** 10.3390/cells12121607

**Published:** 2023-06-12

**Authors:** Dijana Stojanovic, Miodrag Stojanovic, Jelena Milenkovic, Aleksandra Velickov, Aleksandra Ignjatovic, Maja Milojkovic

**Affiliations:** 1Department of Pathophysiology, Faculty of Medicine, University of Nis, 18000 Nis, Serbia; 2Department of Medical Statistics and Informatics, Faculty of Medicine, University of Nis, 18000 Nis, Serbia; 3Center of Informatics and Biostatistics in Healthcare, Institute for Public Health, 18000 Nis, Serbia; 4Department of Histology and Embryology, Faculty of Medicine, University of Nis, 18000 Nis, Serbia

**Keywords:** renalase, mitochondrial dysfunction, cardiac disease, sirtuins, apoptosis, oxidative stress, MAPK, NAD^+^

## Abstract

The cellular mechanisms and signaling network that guide the cardiac disease pathophysiology are inextricably intertwined, which explains the current scarcity of effective therapy and to date remains the greatest challenge in state-of-the-art cardiovascular medicine. Accordingly, a novel concept has emerged in which cardiomyocytes are the centerpiece of therapeutic targeting, with dysregulated mitochondria as a critical point of intervention. Mitochondrial dysfunction pluralism seeks a multi-faceted molecule, such as renalase, to simultaneously combat the pathophysiologic heterogeneity of mitochondria-induced cardiomyocyte injury. This review provides some original perspectives and, for the first time, discusses the functionality spectrum of renalase for mitochondrial dysfunction improvement within cardiac disease, including its ability to preserve mitochondrial integrity and dynamics by suppressing mitochondrial ΔΨm collapse; overall ATP content amelioration; a rise of mtDNA copy numbers; upregulation of mitochondrial genes involved in oxidative phosphorylation and cellular vitality promotion; mitochondrial fission inhibition; NAD^+^ supplementation; sirtuin upregulation; and anti-oxidant, anti-apoptotic, and anti-inflammatory traits. If verified that renalase, due to its multi-faceted nature, behaves like the “guardian of mitochondria” by thwarting pernicious mitochondrial dysfunction effects and exerting therapeutic potential to target mitochondrial abnormalities in failing hearts, it may provide large-scale benefits for cardiac disease patients, regardless of the underlying causes.

## 1. Introduction

Cardiac disease (CD) remains a significant cause of the global disease burden in terms of disability, premature mortality, and spiraling health care costs [1,2,3,4]. It should be noted that 6.2 million deaths in 2019 in the population between 30 and 70 years of age are attributed to CD [1]. Moreover, prevalent CD cases nearly doubled (271 to 523 million), within a two-decade-long period (1990–2019), including an increased number of deaths from 12.1 million to 18.6 million [1]. More ominous data report that the age-standardized rate of CD has begun to rise in high-income countries, where it was hitherto decreasing [1,4]. At the same time, its burden continues to increase for almost all populations outside high-income countries [1], with an annual prediction of 24 million deaths by 2030 [4], accounting for 2–3% of the total global healthcare expense [5]. Despite the involvement of evidence-based therapy, survival rates and quality of life have only modestly improved over the last two decades [3,5]. Therefore, the current targeting of neurohumoral over-activation calls for greatly needed reinforcement to identify the intrinsic mechanisms critically involved in CD initiation and deterioration [5].

The cellular mechanisms and signaling networks that guide the CD pathophysiology are inextricably intertwined, which results in the scarcity of effective therapy. A newly emerging concept in which cardiomyocytes have become the centerpiece of therapeutic targeting along with mitochondria as the critical points for intervention seems alluring and beckons an invitation for more comprehensive scrutiny. A wealth of ongoing research [5,6,7,8,9,10,11,12,13,14,15,16,17] indicates that correcting dysfunctional mitochondria may prove beneficial for myocardial dysfunction improvement within the range of cardiac pathology. For instance, dysfunctional mitochondria do not produce an adequate supply of ATP, altering the ability of cardiomyocytes to produce energy. They also represent the central origin of reactive oxygen species (ROS), as well as their first target, thus perpetuating a wide-ranging role in cardiomyocyte death. A comprehensive appreciation exists for an innovative molecule, a multi-faceted renalase, which may be capable of simultaneously targeting several detrimental pathways of mitochondria-induced cardiomyocyte deterioration and prove to be relevant for treatment of CD.

This review summarizes the most recent knowledge and critically discusses the diversity of renalase mechanisms, such as nicotinamide adenine dinucleotide (NAD^+^) supplementation and sirtuin upregulation, as well as its anti-oxidant, anti-apoptotic, and anti-inflammatory effects, particularly with respect to mitochondrial dysfunction and potential CD improvement. It provides some original perspectives regarding the renalase/mitochondrial causal relationship and, based on the data gained on preclinical research, develops the concept that renalase, due to its functional diversity, behaves as the “guardian of mitochondria”, and its application may mitigate mitochondrial dysfunction and CD progression. Regarding the suggested concept, this review may be used as an accurate reference to establish additional novel pathways through which mitochondrial dysfunction may be treated, serving as a knowledge base for the design of innovative CD preclinical models and the potential role of renalase. The initial discussion will concisely address key mitochondrial dysfunction pathways and the pathophysiologic characteristics of renalase; their proposed relationship will be discussed throughout the review.

## 2. Mitochondrial Pathophysiology within the Concept of Cardiac Disease

It is possible that the target point for the best CD treatment is a viable cardiomyocyte and its intrinsic activities [5]. Given that mitochondria comprise 25–30% of cardiomyocytes [5,8,10,18] and that 95% of ATP consumed by the heart is produced by mitochondria [10,11,12,13], it may be foreseen that any disruption in their function results in a diminished competency to produce ATP followed by detrimental responses, such as mitochondrial (mt)ROS overproduction, apoptosis promotion, and sterile inflammation. In addition, mitochondrial dysfunction amelioration upholds beneficial outcomes that extend beyond solely improved cardiac function, given that skeletal muscles in CD patients also exhibit mitochondrial dysfunction and, thus, contribute to exercise intolerance [19]. Mitochondrial dysfunction is also present in patients with insulin resistance [20] and kidney disease [21], which are typical features of CD pathophysiology. 

Mitochondria are dynamic structures enclosed by two membranes and a negatively charged matrix [5,6,7,8,9,10,11,16,17,18]. The interface of the inner mitochondrial membrane (IMM) assembles a non-specific channel characterized by limited permeability known as the mitochondrial permeability transition pore (mtPTP), which serves as the regulator of mitochondrial Ca^2+^ homeostasis [11]. Unlike other organelles, mitochondria have a DNA (mtDNA) genetic code that differs from cardiomyocyte DNA [5,10] and which codes all subunits of the electron transport chain (ETC) [16,18]. The primary function of the cardiac mitochondria is, through oxidative phosphorylation, to provide ATP that is essentially required for heart contraction and relaxation. This process occurs in the IMM, which is impassable for the majority of ions and small molecules where the reducing equivalents, such as nicotinamide adenine dinucleotide (NADH) and flavin adenine dinucleotide (FADH_2_) generated in the Krebs cycle and β-oxidation, transfer electrons through the ETC [5,10,13,16]. The efflux of protons is followed by the electronic flow on ETC (proton electrochemical gradient). It generates a mitochondrial membrane potential (ΔΨm) which is used to convert ADP to ATP using the ATP synthase. Accordingly, the wholesome integrities of the IMM and the mitochondrial membrane potential are critical points for mitochondrial function maintenance [13,15,16]. The generation of ATP through the respiratory chain is accompanied by ROS production, particularly ·$O_{2}^{-}$, OH, and H_2_O_2_. These oxygen species are short-lived and highly reactive molecules, which may pose harmful effects for the metabolic function of cardiomyocytes by inducing lipid peroxidation, irreversible DNA and RNA damage, protein carbonylation, and imbalanced cellular redox, thus leading to mitochondrial deterioration and apoptosis initiation as well as other types of programmed cell death [9,15,16]. Additionally, a competent multilayer detoxifying system has evolved (superoxide dismutase (SOD), catalase, peroxidase) to compensate for oxygen-mediated mitochondrial damage [9,10,13,14,15,16]. However, the efflux of superoxide from the mitochondrial ETC quickly experiences dismutation by SOD to molecular oxygen and water, making cardiac tissue highly adaptable to redox changes for maintaining healthy contractility [9,15,16]. During increased ROS production (oxidative stress (OS)), mitochondria are capable of introducing particular signals to preserve membrane polarization, adenine nucleotide levels, further ROS generation, excitation/contraction coupling, and the opening/closing of mtPTP to restore myocardial tissue upon injury [9]. The oxidative metabolism of cardiac mitochondria goes far beyond ATP generation. They are significantly involved in cell signaling regulation via redox state modulation, supplying metabolites for protein synthesis and modification and providing biochemical reaction cofactors and ligands for signaling transduction [5,6,13,14,15,16,22].

Mitochondria are also involved in the regulation of innate immune responses, either through direct or indirect activation, via inflammatory outcome modulation [18]. In response to cellular stressors and loss of homeostasis, a disrupted permeability of the mitochondrial membrane occurs, thus allowing the release of mitochondrial insides, i.e., damage-associated molecular patterns or DAMPs (ROS, mtDNA, and other mitochondrial components), which are recognized by the innate immunity receptors (toll-like receptors, the cytosolic nucleotide-binding oligomerization domain-like receptors, NLRs) which trigger the immune response [18]. Some NLRs, upon induction, compose a multi-protein complex named “inflammasome” to induce the cleavage and activation of IL-1β and IL-18 by caspase-1 and promote pro-inflammatory actions. Mitochondria may initiate and regulate the NLRP3 inflammasomes [8,18] to further disrupt mitochondrial homeostasis, trigger NLRP3 de-ubiquitination and the release of mitochondrially derived molecules and destroy mtDNA [9]. Taken together, the dynamic mechanisms that mediate mtROS overproduction, mutated mtDNA accumulation, and mtDAMP release are inevitably tangled and behave as stimuli for a cascade of pro-inflammatory malaise which lead to a perpetual cross-linking of different cardiac cells, accumulated inflammatory cells, and diverse signaling pathways, thus resulting in maladaptive alterations of the heart structure and functions. Following the aforementioned, certain cardiomyocyte stress settings initiate the opening of the mPT, which induce the decline of mitochondrial membrane potential, resulting in insufficient ATP production, and the release of mitochondrial pro-apoptotic proteins, triggering a complex network of cardiomyocyte death (mitochondria-dependent necrosis and apoptosis) [8,11,13]. The initiation of mitochondria-mediated apoptosis occurs through the interconnection of an intrinsic and extrinsic “death” network, whereby the intrinsic pathway induces mitochondrial outer membrane permeabilization, followed by cytochrome C release, formation of the apoptosome, activation of pro-caspase-9, and subsequent cleavage of caspase-3 and caspase-7 [15]. The intrinsic apoptosis network is mediated by the B-cell lymphoma 2 (Bcl-2) family which regulates the opening of the mitochondrial outer membrane in response to either death receptors or various cellular and/or mitochondrial stresses [15,16]. If robustly damaged, mitochondria operate as cellular sensors for initiation of the intrinsic pathways, leading to “a point of no return” and progressive loss of cardiomyocytes [12].

Beyond anti-oxidative, anti-apoptotic, and anti-inflammatory protection, additional mechanisms crucial to maintaining mitochondrial homeostasis and cellular survival [5,10,11,12,13,14,15,16] are the regulated processes of mitochondrial quality control in the form of mitochondrial biogenesis, mitochondrial dynamics (fission, fusion), and autophagy (mitophagy). These systems are dysregulated in some CD studies [6,9,11,12,23] and may potentially be a point of therapy targeting [6,9,10,11,12]. Mitochondrial biogenesis comprises an increased number of functional mitochondria, improved mtDNA repair, and an adequate synthesis of mitochondrial proteins [9,10]. Accordingly, dysregulated mitochondrial biogenesis likely relates to increased mitochondrial protein acetylation, followed by impaired cardiac energy metabolism, increased mtROS generation, mPTP opening, and DAMP release, thus inducing inflammatory response [9,13,15,18]. The process of protein hyperacetylation may be regulated by sirtuins (NAD^+^-dependent deacetylases), whereby the diminished cellular NAD^+^ levels and reduced NAD^+^/NADH ratio are strongly associated with mitochondrial protein hyperacetylation and impaired function [5,13,15]. The constant and balanced mitochondrial processes of fission and fusion refer to mitochondrial dynamics [9,10,11,15,23], which are relevant for mitochondrial self-renewal to form new mitochondria (fission) and/or to restitute damaged mitochondria (fusion). These processes are under the supervision of dynamin-related guanosine triphosphatase (GTP-ase) proteins [10,11,15], such as dynamic-related protein (Drp)-1, identified as the predominant fission protein. However, Drp-1 interacts with some receptors located in the outer mitochondrial membrane, including the mitochondria fission factor (Mff) known as fission protein 1 (Fis1) and the mitochondrial dynamic proteins 49 and 51 (MiD49/51) [10,15], to promote the recruitment of Drp-1 to mitochondria. Nevertheless, disruption of the fission/fusion balance results in a loss of pH gradient, the collapse of ΔΨm, mtDNA injury, permanent mtPTP opening, impaired cardiac bioenergetics, and oxidative damage and represents an early event in mitochondria-dependent apoptosis [9,11,15]. Mitophagy refers to a specific form of autophagy for the removal of increased mitochondrial damage associated with OS and impaired quality control, [9,10,13,15] and is initiated in settings related to the impairment of mitochondrial membrane potential.

All things considered, mitochondrial dysfunction contributes to abundant cardiac pathologies: ischemia-related disorders, atherosclerosis, heart failure, and metabolic disease [6,7,8,9,10,11,12,13,14,15,16,17,19,22,23,24], whereby its pharmacological restoration emerges as an intriguing therapeutic avenue. Some of the mitochondria-mediated targets in the chronically stressed heart may be the pathway to balancing NAD^+^/NADH redox with NAD^+^ precursors to regulate substrate oxidation, NAD^+^-dependent protein deacetylation, and mitochondria-induced inflammatory response, thus enabling mitochondrial biogenesis preservation [13]. The mtROS hyperproduction and sustained low-grade inflammatory response with its consequences (mitochondria-mediated apoptosis and inflammasome activation) also represent challenging targets. These pathways are outlined in the sections that follow to acknowledge a potential relationship between them and the innovative molecule renalase and to provide scientific plausibility for the proposition that their concurrent targeting by renalase may be potentially effective for mitochondrial dysfunction improvement. An amount of in vitro and in vivo evidence exists that renalase functions as a key regulator of mitochondrial wealth, contributing to the pool of healthy mitochondria through mechanisms such as NAD^+^ supplementation and sirtuin activation, mitochondrial fission improvement, silencing of ROS production, and an innate inflammatory response. However, current and future research focusing on CD pathophysiology warrants the validation of the effectiveness of renalase as a force in improving mitochondrial morphology, metabolism, and cardiac function.

## 3. The Molecular Signature of Renalase

The research on mechanisms and molecules implicated in maintaining renal and cardiovascular health resulted in the discovery of a novel protein, named renalase, upon its abundant renal expression [25,26,27]. The knowledge that end-stage chronic kidney disease represents an established risk factor for the development of cardiac complications (cardiac hypertrophy and fibrosis) raised the possibility of a hitherto unrecognized molecule providing an interaction between the heart and the kidneys. After a thorough analysis of 114 candidate genes encoding new secretory proteins, one clone was discovered to be preferentially expressed in the proximal tubules and cardiomyocytes [25,26,27]. Subsequent research demonstrated that the tissue expression of renalase, to a varying extent, exists in the liver [28,29,30], small intestines [31], pancreas [32,33], brain [34], skeletal muscles [35,36,37,38], the cornea [39], the placenta [40], and in various cancers [33,41,42,43,44], thus indicating a molecule of a multi-faceted nature that likely preserves homeostasis in general.

The primary structure of renalase parallels that of monoamine oxidase (MAO)-A and MAO-B, sharing 14% of its amino acid identity and obtaining a flavin-adenine dinucleotide (FAD) for its catalytic activity [25,26,27]. The renalase gene is located in chromosome 10 (at q23.31), containing approximately 300,000 nucleotides across 11 exons [25,26,27,44]. It circulates within the blood at a concentration of approximately 5 µg/mL [44] and is present in two major isoforms (1 and 2) [45,46]. It has been demonstrated that two more isoforms (3 and 4) markedly shorten amine oxidase domains [44,45,46] and lack oxidase activity. The presence of these variances [27] is accountable for not fully elucidating renalase responses, molecular pathways, and the particular cellular effectors with which renalase interacts. Both renalase isoforms (1 and 2) have been revealed to comprise essential functional domains: N-terminal signal sequences, a FAD-binding domain, and the Rp-220 site [44,46,47]. In terms of the feasible mitochondrial protection of renalase, its enzymatic NAD^+^ oxidase domain, located in the N-terminal region, and the Rp-200 peptide sequence, essential for renalase interaction with cell receptors in terms of anti-inflammatory response and cell survival, are of particular interest [44,47,48]. It has been demonstrated that the nature of renalase is presumably two-fold: acting as an intracellular catalytically effective flavoprotein (EC 1.6.3.5 dihydro-NAD(P): oxygen oxidoreductase) and as a cytokine, thus providing signaling capabilities [44,47]. 

The intracellular pathways and allocation of renalase remain elusive and may be tissue- or isoform-dependent, which are likely influenced by various local and systemic determinants [44]. Several lines of evidence confirm transcriptional factors responsible for the upregulation of renalase gene expression: NF-κB, HIF-1α, STAT3, TNF-α, ZBP89, and Sp-1 [31,49,50,51,52,53], several of which are accountable for inflammatory actions. However, considering that renalase has been shown to operate as a NADH oxidase [44] that oxidizes and epimerizes α-NAD(P)H [53,54,55], the most prominent response of intracellular renalase is the oxidative metabolism of NAD(P)H. Two other substrates that have been demonstrated for renalase oxidation are 2- and 6-dihydroNAD(P), whereby the isomeric forms of β-NAD(P)H occur by nonspecific reduction in β-NAD(P)^+^ or by tautomerization of β-NAD(P)H (4-dihydroNAD(P)) [55]. Such results imply that to abolish the perils they represent for normal respiratory activity, renalase is metabolically active by oxidizing the isomeric NAD(P)H molecules to β-NAD(P)^+^ [55]. The realization that renalase converts 2 (or 6) NADH (metabolically inert) into active 4-NAD^+^ provides proof that intracellular renalase functions as a point of NAD^+^ supply by significantly modulating its cellular levels and regulating mitochondrial energy production, thus identifying renalase as a cellular “housekeeper”. Given that NAD^+^ represents a molecule essential for energy metabolism and DNA repair, including its role as a coenzyme in sirtuin metabolism [10,56,57,58], it could be hypothesized that renalase may be indirectly included in different pathways of sirtuin functions, including mitophagy.

Apart from its enzymatic nature as an NADH oxidase [59,60], subsequent research endorses renalase as a signaling molecule, which, upon binding to its receptor, i.e., the plasma membrane Ca^2+^-ATP-ase4b (PMCA4b) [44,61,62], sets in motion an intracellular prosurvival signaling network. PMCA4b is identified as a broadly scattered low-capacity plasma-membrane Ca^2+^ export pump that provides the numerous biological effects of renalase [44,48]. The signaling elicited by renalase enables the regulation of multiple intracellular-regulated protein kinase pathways: phosphatidylinositol 3-kinase/protein kinase B (PI3K/Akt), extracellular signal-regulated kinase (ERK)1/2, p38 mitogen-activated kinase (p38), B-cell lymphoma 2, and c-Jun N-terminal kinase (JNK) [44,48,61,62]. Furthermore, renalase evokes important downstream pathways: Ras/Raf/MEK/ERK, p38, NOS, NF-κB, PI3K/Akt, and cAMP [44,61,62] to provide a cytoprotective response, as presented in Figure 1. Finally, renalase can hinder inflammasome activation and the release of IL-1β [32,44,62]. Taken together, renalase favors significant cytoprotection related to inflammation, oxidative stress, and apoptosis [28,29,30,31,32,37,38,39,50,59,62], whereby recent research also discloses its potential in diverse cardiac [63,64,65,66,67,68,69,70,71,72,73,74] and kidney pathologies [75,76,77,78] and in the pathophysiology of COVID-19 [79,80]. It will be captivating to conduct future scrutiny toward an excellent comprehension of renalase pleiotropy.

## 4. The Network of Renalase/NAD^+^/Sirtuins in Regard to Mitochondrial Dysfunction within Cardiac Disease

Sufficient evidence has shown that augmented acetylation of several mitochondrial proteins (fatty acid oxidation enzymes, pyruvate dehydrogenase, or ETC proteins) leads to significantly altered mitochondrial structure and function and occurs during CD progression [13,81,82,83]. Acetyl proteomics acknowledges a substantial hyperacetylation of lysine within mitochondria in the early stages of heart failure in mouse models and in the terminal stages of a failing human heart [81,83]. Mitochondrial protein hyperacetylation diminishes the activity of succinate dehydrogenase, pyruvate dehydrogenase, and ATP synthase, while the hyperacetylation of oligomycin sensitivity-conferring protein raises the sensitivity of the mtPTP opening [13,82]. Protein hyperacetylation hinders the transition of NADH from the cytosol to the mitochondria, leaving the cellular redox state and ATP production impaired, which is observed during CD progression [83]. A current hypothesis suggests that protein hyperacetylation in the mitochondria of cardiomyocytes is initiated by augmented acyl-CoA availability and maintained by a NAD^+^/NADH redox imbalance [13,84,85,86]. For better clarification, during the process of fatty acid β-oxidation, NAD^+^ serves as a hydride receiver which is reduced to NADH and is oxidized afterwards to NAD^+^, generating ATP. The NAD^+^/NADH ratio is also required for cellular metabolism and mitochondrial oxidative phosphorylation [86,87,88,89]. Settings of cardiac ischemia and myocardial infarction affect NAD^+^ synthesis, ensuing in the depleted NAD^+^ content of cardiomyocyte and accompanied by overall mitochondrial dysfunction [84,85,86,87,88,89,90,91]. Accordingly, in advanced age, myocardial NAD^+^ levels are diminished and associated with a decline in mitochondrial function [87,88,89]. During the CD progression, cardiomyocytes use glycolytic and ketone body oxidation instead of fatty acid β-oxidation, resulting in a more depleted NAD^+^/NADH ratio, which further affects cardiac function. If sustained, the setting of the decreased NAD+ content or/and NAD^+^/NADH ratio increases ROS generation, mitochondrial pore opening, antioxidative protection weakening, Ca^2+^ overload, and augmented inflammation [13], thus leading mitochondria to the point of promoting a programmed cell death. Moreover, the decreased NAD^+^ or NAD^+^/NADH ratio may be accountable for the downregulation of proteins that operate as deacetylating enzymes, such as the sirtuin family of protein NAD^+^-dependent deacetylases, resulting in altered mitochondrial protein deacetylation, mitochondrial dysfunction, and impaired energy metabolism [13,86,87,88,89,90]. It may be foreseen that mitochondria-rich tissues such as the heart display a higher NAD^+^ content and yield to its maintenance [86,87,89,90,91]. Along this rationale, evidence documents that decreased bioavailability of NAD^+^ contributes to the development and progression of atherosclerosis, ischemia-reperfusion injury, arrhythmias, hypertension, dilated cardiomyopathy, and heart failure [84,85,86,87,88,89,90,91]. It may be expected that any strategy that is competent in enhancing the level and/or activity of the NAD^+^ metabolome (or NAD^+^/NADH ratio) followed by sirtuin activation and improved cellular processes may mitigate a wide range of cardiac pathologies [57,58,84,85,86,87,88,89,90,91].

As hitherto explained, the common point for sirtuin (SIRTs) regulation is their cofactor NAD^+^ [58,84,85,86,87,88,89,90]; in their presence, SIRTs catalyze the deacetylation of histone, some transcriptional regulators in the nucleus, and several proteins in the mitochondria and cytosol [86,87,88], thus yielding protection against aging, senescence, hypoxia, metabolic diseases, and sustained inflammation [85,86,87,88,89]. In this way, SIRTs improve the cardiomyocyte competence for metabolic adaptation and operate as a compelling “cardiomyocyte protector” [87,88,92,93,94,95,96,97,98]. Sirtuins maintain metabolic homeostasis, decrease cellular damage, and silence pro-inflammatory responses in a variety of cardiac pathologies: atherosclerosis, myocardial infarction, cardiac hypertrophy, and heart failure [86,87,92,93,94,95,96,97,98]. Comparatively, the experimentally induced settings of cardiovascular risk factors, i.e., hypercholesterolemia, obesity, and type 2 diabetes mellitus, exert significant amelioration upon pharmacological modulation of SIRT activity [87]. The most relevant SIRTs within the CD pathophysiology are SIRT1, localized in the nucleus and cytosol, also involved in mitochondrial biology, and SIRT3, primarily localized in the mitochondria [84,86,92,93,94,95,96,97,98,99].

Sirtuin 1 controls the acetylation of p53, which is deacetylated and repressed upon DNA damage or oxidative stress, thus leading to impaired apoptosis and mitochondrial dysfunction [86,92,93,98], and deacetylates PGC-1α, a transcriptional regulator of mitochondrial biogenesis and activity, to activate downstream pathways for mitochondrial gene expression [99,100,101,102]. Moreover, SIRT1 mediates the deacetylation of forkhead box O (FOXO) transcription factors, which are essentially involved in lipid and glucose metabolism regulation and stress response [103,104]. It also directs FOXO to particular targets to indirectly regulate mtROS production and a cellular anti-oxidant expression [93,103,104]. Pharmacologically induced SIRT1 activity has been shown to increase the density of hepatic LDL-cholesterol receptors, followed by reduced LDL-cholesterol plasma concentration, to confer atheroprotective effects [87,105]. Regarding diverse cardiac stresses (pressure overload, nutrient starvation, exercise, and acute ischemia), SIRT1 expression has been evidenced to increase, while its levels are downregulated during ischemia/reperfusion (I/R) injury [106,107]. Along these lines, experimentally induced SIRT1 activation alleviates cardiac I/R injury via the upregulation of ERK phosphorylation and downregulation of p38 and JNK expression [106,107], while NAD^+^ supplementation recovers the cardiac phenotype in a SIRT1-dependent mechanism [107].

Sirtuin 3 is localized in the mitochondrial matrix and, at some point, in the nucleus and cytoplasm, yielding the most robust deacetylase response in mitochondria [94,95,96,97]. Beyond the deacetylation of histones, SIRT3 deacetylates enzymes involved in fatty acid and glucose oxidation [85,86,95,97] and protein subunits of the ETC, including the enzymes required for oxidative defense and mitochondrial homeostasis. Additional SIRT3-mediated functions may be a nuclear translocation of FOXO3a (SIRT3-FOXO3a pathway), similar to that of SIRT1, to induce the specific sets of genes, including ROS scavengers, autophagy effectors, DNA repair, and gluconeogenic enzymes [95,108,109]. Sirtuin 3, via the modulation of apoptosis, synergistically with its ability to activate or inhibit autophagy, retains far-reaching protection, thus enabling cells to endorse the most appropriate stress response and maintain mitochondrial safety [108,109]. Some mitochondrial disturbances, such as impaired oxidative phosphorylation, inadequate ATP production, oxidative burst, increased sensitivity for mPTP induction, and critical pathways in mitochondrial dysfunction-induced diseases, may be regulated in a SIRT3-dependent manner [95]. A wealth of evidence suggests that the downregulation of SIRT3 represents a causative factor in cardiac hypertrophy and fibrosis, followed by heart failure [94,95,96,97]. For instance, there is evidence that SIRT3 is a negative regulator of cardiac hypertrophy via the activation of FOXO3a-dependent anti-oxidants and by impeding mtROS-mediated Ras activation and downstream MAPK/ERK and PI3K/Akt signaling [108,109]. It has also been involved in the pathophysiology of coronary atherosclerosis, ischemic heart disease, and myocardial infarction, while SIRT3 downregulation correlates with poor cardiac recovery after ischemia [94,95,96,97]. The lack of SIRT3 expression promotes cardiac lipotoxicity and hypertension, contributing to endothelial dysfunction, vascular hypertrophy, inflammation, and irreversible cardiac damage [96,97,110].

Within that context, admittedly based on a piece of scant evidence, recombinant renalase supplementation in animal models with acute cellular injuries results in the preservation of mitochondrial homeostasis, evidenced by hindered mitochondrial fission, preserved mitochondrial dynamics, and improved mitochondrial function in general, followed by oxidative stress alleviation and apoptosis suppression. These renalase-mediated responses are recovered in a sirtuin-dependent manner, presumably by boosting NAD^+^ cellular content. The initial hypothesis that linked renalase to the oxidative metabolism of NAD(P)H and raised the awareness that renalase may be a tool for cardiomyocyte energy metabolism preservation [44,53,54,55] was established upon evidence that renalase deficiency results in significant cardiomyocyte necrosis and mild ventricular hypertrophy [59]. The degree of infarct size is three times more severe than the controls, suggesting that renalase deficiency predisposes toward aggravated ischemic cardiac damage. The same study also demonstrated that reperfusion of the ischemic hearts of renalase knockout animals with a buffer containing renalase or glutathione synthase significantly rescues the cardiac phenotype and improves cardiac function. Nonetheless, a lack of renalase alters the NAD^+^/NADH ratio of the cardiomyocytes, together with decreased plasma and tissue NADH oxidase activity. At the same time, in vitro testing confirms that renalase obtains substantial plasma NADH oxidase activity, potentially contributing to the regulation of NAD^+^ levels [59]. The evidence of the ability of renalase to sustain and, when needed, recover NAD^+^ levels, potentially regulating cellular redox metabolism, creates a knowledge base for renalase scrutiny related to the members of the sirtuin (NAD^+^-dependent deacetylase) family. In light of this hypothesis, it was documented that renalase protects hepatic cells by alleviating oxidative stress and boosting NAD^+^ levels, followed by significantly upregulated SIRT1 activity [29]. Renalase behavior may be context-dependent, contingent on the type of cellular damage: its expression is downregulated in chronic settings and upregulated in the settings of ischemia/reperfusion and increased oxidative stress. Renalase administration effectively alleviates the degree of necrosis and attenuates apoptosis by suppressing cytochrome C liberation and Bax expression and increasing Bcl-2 expression. In particular, renalase pretreatment prevents mitochondria ΔΨm collapse by increasing overall ATP content and mtDNA copy numbers and suppressing mitochondrial translocation of the mitochondrial fission-related protein Drp-1. This preserves mitochondrial network morphology and provides substantial mitochondrial quality control. Finally, renalase positively correlates with SIRT1 expression and activity, considering that in the setting of renalase administration, SIRT1 activity and expression are increased and vice versa. The intriguing hypothesis of the renalase/SIRT1 axis is postulated, concluding that renalase operates as NAD^+^ booster followed by SIRT1 upregulation, promotion of depleted acetylation levels of p53, negative regulation of Bax expression, and upgraded Bcl-2 expression. The evidence suggests that renalase prevents mitochondrial fission, resulting in sustained mitochondrial dynamics and homeostasis and hypothetically mediates mitochondria bioenergetic adaptation. Fundamentally, this corresponds to the suggested mechanisms of mitochondrial dysfunction in CD pathophysiology. This recognition is reinforced in a similar preclinical model of acute kidney injury [111], where renalase pretreatment lessens the number of apoptotic cells and reduces Bax upregulation and cleaved caspase-3 and upgrades Bcl-2. Accordingly, renalase treatment facilitates the expression of SIRT3, hinders mitochondrial fragmentation, and abolishes the possibility of a ROS burst, thereby preventing mitochondrial fission and mitochondrial dysfunction. It is evidenced that the modulatory effects of renalase on mitochondrial dynamics are accomplished by its competence in enhancing the phosphorylation of Drp-1 at serine 637 and impeding the translocation and accumulation of Drp-1 to mitochondria. Collectively, renalase pretreatment significantly reduces ROS production, apoptosis, and mitochondrial fission by preventing the downregulation of SIRT3 and recovering its expression (SIRT3-dependent manner). It may be hypothesized that renalase supplementation, in a likely demeanor, may benefit failing hearts, given the wealth of evidence that cardiac and renal diseases share some pathomechanisms [86] and that modulating the NAD^+^/sirtuin axis, potentially by renalase, may be a clinically relevant approach for developing new therapies. If proven that renalase exerts the sirtuin-positive regulatory effects in cardiomyocytes, similar to those evidenced in kidneys or hepatocytes, it may serve to ameliorate the outcome for CD patients. The most enthralling hypothesis, without evidence so far, suggests that renalase potentially initiates sirtuin-regulated mitophagy to hamper myocardial remodeling in acute myocardial infarction. An illustration of the proposed renalase and sirtuin interplay is depicted in Figure 1, and the summary of evidence-based renalase traits that may prove applicable in cardiac disease pathophysiology is presented in Table 1. 

NF-κB, nuclear factor κB; HIF-1α: hypoxia-inducible factor-1α; NOX 2/4: nicotinamide adenine dinucleotide phosphate oxidases 2/4; Bax: Bcl-2-associated X protein; Bcl-2: B-cell lymphoma 2; NAD^+^: nicotinamide adenine dinucleotide; NAD(P)H: nicotinamide adenine dinucleotide phosphate; PI3K/Akt: phosphatidylinositol 3-kinase/protein kinase B; MAPKs: mitogen-activated protein kinases; JNK: c-Jun N-terminal kinase; AKI: acute kidney injury; ROS: reactive oxygen species; Drp1: dynamin-related protein; SIRT, sirtuin; MDA: malondialdehyde; SOD: superoxide dismutase; EMT: epithelial-mesenchymal transition; α-SMA: α-smooth muscle actin; MCP-1: monocyte chemoattractant protein-1; TIMP-1: tissue inhibitor of metalloproteinase-1; MMP-1: matrix metalloproteinase-1; ERK1/2: extracellular regulated protein kinases 1/2; CKD; chronic kidney disease; TLR: toll-like receptor; MIP-2, macrophage inflammatory protein 2.

Based on the experimentally acquired knowledge of renalase pathophysiology, the hypothesis exists that renalase, NAD^+^, and sirtuins (1 and 3) share some common aspects to the point that the behavior of renalase, in terms of mitochondrial morphology and dynamics, may be multi-faceted. Indirectly via NAD^+^ supplementation and sirtuin (1 and 3) upregulation and directly by abrogating oxidative stress and apoptosis, renalase operates as a “guardian of mitochondria”. Such perception appears thought-provoking, considering that mitochondria-rich cardiomyocytes, being metabolically active cells, may benefit from the versatility of renalase in a diseased setting. Nevertheless, regarding the postulated renalase/NAD^+^/sirtuin axis, ample gaps of knowledge have yet to be addressed for a better understanding of the exact molecular pathways underlying the proposed axis and whether renalase may maintain mitochondrial homeostasis in the CD setting. 

## 5. The Interplay of Renalase and ROS in Regard to Mitochondrial Dysfunction in Cardiac Disease

When maintained at a low level, ROS acquire some physiological roles in various processes exerting a beneficial response in cardiac mitochondrial quality control [5]. However, the setting of their excessive production and/or decreased clearance (oxidative stress), significantly contributes to cell and organ damage [5,13,16,17,121,122,123,124]. Mitochondria represent the main source of ROS within the cardiomyocytes, specifically along the IMM and the mitochondrial matrix by the ETC components. The process of mitochondrial respiration generates mtROS, with the most reactive superoxide anion (·$O_{2}^{-}$) among them, which is highly competent for further oxidation processes and the de novo generation of a great number of free radicals (H_2_O_2_, peroxynitrite, and hypochlorous acid) in a mitochondrial matrix concentration 5 to 10-fold higher than that of the cytosol or nucleus [16]. For mtROS detoxication, SOD efficiently dismutases superoxide to H_2_O_2_, with SOD2 as the primary mitochondrial isoform [13,16]. At the same time, mitochondrial H_2_O_2_ is removed via the antioxidative system of peroxiredoxin and glutathione peroxidase [13] and requires NADPH as an electron donor [16,121]. It is noteworthy that NADPH is of substantial value for mtROS scavenger enzymes to maintain the pool of reduced glutathione, glutaredoxin, and thioredoxin, whereby the decline of major enzymes for NADPH generation and/or recycling subsides the cell competence to detoxify mtROS [121]. Due to the lack of efficiency of ROS scavengers or their outpacing by augmented mtROS production, this phenomenon results in increased mtROS and associated reactive intermediate levels, capable of damaging cell proteins and lipids to elicit a cell-death cascade and triggering the disruption of the cellular energy network [5]. The augmented mtROS generation, followed by downstream ROS-mediated damage, has been acknowledged in CD animal models and disclosed as the major pathogenic factor in CD progression [5,13,16,121,122,123]. For instance, increased mtROS, through the modification of proteins important to excitation/contraction coupling, alter cardiomyocyte electrophysiology and their contractile function [122]. Oxidative stress alters myofilament calcium sensitivity, impairing the activity of the sarcoplasmic reticulum Ca^2+^-adenosine triphosphatase (SERCA) and promotes the proliferation of cardiac fibroblast and matrix metalloproteinases, thus leading to cardiac fibrosis progression [122]. As signaling molecules, mtROS regulate the activity of NF-κB, including the transcriptional expression of the related genes and endothelin production through the RAS/RAF/ERK pathway, thus eliciting adverse cardiac remodeling [123]. Moreover, mtROS promote oxidative modification of some mitochondrial enzymes, and, thus, mitochondrial dysregulation, while the mtROS-induced mPTP opening induces cell death [13,16,125]. A strong correlation exists between mtROS overproduction, mtDNA damage, and altered mitochondrial biogenesis [13], while mtROS-induced mtDNA leakage likely results in the initiation of an inflammatory response with its consequences [13,18,121,122,123,124]. Moreover, mtROS directly modulate the expression and the functioning of microRNA (small non-coding RNA included in posttranscriptional regulation of protein expression), several of which have been documented to modulate mitochondrial function, and ultimately resulting in cardiomyocyte proteome remodeling and a metabolic shift, thus contributing to CD [126]. In addition to their direct cellular damage or induced downstream signaling, mtROS enhance the generation of other types of reactive molecules (aldehydes), trigger the “mtROS-induced ROS” process, and mtROS-mediated paracrine signalization [13,16,17,121,122,123,124,127]. This phenomenon explains, at least in part, why the exogenous inhibition of ROS production or supplementation of exogenous anti-oxidants did not favor an adequate level of beneficial effects [1,5,121]. The final remark may be that the increased oxidative stress of cardiomyocytes, whether the cause or the consequence of mitochondrial dysfunction (presumably both), represents a significant demarcation point for the deterioration of the failing heart before final ATP depletion and should be targeted in a multifaceted manner.

The introductory observation that OS triggers renalase expression in the cell culture [27] provided supremacy for the following elucidation on renalase pathophysiology, aiming to establish the mechanisms for its anti-oxidation traits. It has been suggested that renalase functions as a sensitive ROS-responsive gene given that, during hepatic ischemia/reperfusion injury, its levels have been shown to significantly increase in cells and circulation [28]. Nonetheless, upon cell culture incubation with H_2_O_2_ (increasing concentrations for various periods), the OS-induced mRNA expression of renalase increases proportionally and positively correlates with the time of incubation, while increased serum levels of renalase positively correlate with the degree of necrosis [28]. This observation implies that renalase immediately and competently responds to an oxidative burst and that its feedback is likely dependent upon the intensity, degree of severity, and duration of the oxidative injury. Additionally, evidence shows that H_2_O_2_-induced renalase expression may be successfully abolished by anti-oxidative therapy (the SOD/CAT preincubation). Collectively, the rapid feedback of renalase to an ROS-burst, which may be determined in the blood, its close relation to the OS degree in the extent of tissue damage, and its capacity to respond to the antioxidative agents point to renalase as a captivating biological marker for the severity of oxidative damage.

Regarding the aforementioned data, related research revealed that renalase is, to various levels, involved in OS alleviation and organ damage mitigation, given that a deficit in the renalase gene aggravates oxidative damage, macrophage accumulation, and TGF-β1 expression [30]. Renalase knockout mice are prone to oxidative injury, presumably owing to decreased levels of γ-glutamyl cysteine synthetase catalytic units, which contribute to the maintenance of the intracellular redox balance by glutathione synthesis. Moreover, a strong and positive correlation between renalase and OS was validated in the jejunum and ileum of fasting animals, which exhibit OS-induced intestinal dysfunction and atrophy [31]. Research has evidenced that fasting-induced OS significantly upregulates intestinal renalase expression, induced by NF-κB p65 activation. It is possible that the p65 upregulation induces anti-oxidant gene expression in line with renalase upregulation for the maintenance of anti-oxidative activity. In a cell culture treated with H_2_O_2_, the survival rate appears to be significantly higher upon renalase administration, while renalase overexpression decreases levels of dinitrophenylhydrazine (DNP), a marker of carbonylated protein. Accordingly, the intestinal relative expression of renalase correlates with OS markers, such as DNP and TBARS (thiobarbituric acid reactive substances), which is a marker of lipide peroxidation. The evidence that renalase expression is uniformly increased in the ileum, which is abundant in bacteria that likely make the response sensitive to OS, provides proof that renalase may have a specific cryptic anti-oxidant purpose, resembling the function of glutathione peroxidase. An additional hypothesis was postulated that renalase likely suppresses another mtROS-mediated process, such as the epithelial–mesenchymal transition (EMT), therefore exhibiting further cytoprotection [112]. Namely, upon induction of H_2_O_2_-mediated cellular damage, the evidence indicates increased ROS generation, followed by appropriate changes in MDA and SOD while the cells undergo the processes of EMT followed by fibrosis and validated as the upregulation of α-smooth muscle actin (SMA), collagen I, fibronectin, and downregulation of E-cadherin. However, renalase pretreatment results in a decline in the level of MDA and improved SOD activity and successfully abrogates OS-induced EMT and related fibrosis markers. For better clarification, the EMT refers to a complex cellular process [128,129,130,131,132] during which epithelial cells, stimulated by the signals ROS, proinflammatory cytokines, HIF-1α, TNF-α, and TGF-β1, acquire the phenotype and the behavior of myofibroblasts. In terms of the mechanisms, OS, the inflammatory response, and TGF-β1 signaling [128,129,130] are likely have a pivotal role in EMT initiation, with the aim of augmenting the number of fibroblasts [131,132]. There is evidence that EMT contributes to the advance of cardiac fibrosis following heart injury but also plays a part in atherosclerosis, valvular disease, and fibroelastosis [132]. Therefore, the modulation of mtROS-induced EMT represents a projected target in several cardiac pathologies, which may be possible to achieve upon renalase administration. Moreover, it has been demonstrated that dysfunctional mitochondria-derived ROS regulate TGF-β1-induced EMT, followed by a decline in mitochondrial membrane potential and intracellular glutathione levels, while fibronectin expression (EMT marker) may be hindered by ROS scavengers [133]. The proposed mechanisms resemble a renalase-mediated protection against renal fibrosis, as described; therefore, a theory exists that renalase could weaken the dysregulated mitochondria-derived ROS-induced EMT. The most recent findings [134] also disclose the ability of renalase to hinder EMT progression, promoted by TGF-β1 signaling and implying that renalase-regulating differentially expressed genes are significantly enriched on several pathways, including the PI3K/Akt and the MAPK. Likewise, corresponding research [113] showed that renalase pretreatment ameliorates renal function and histological damage and diminishes cellular necrosis and apoptosis, while oxidative stress is ameliorated through a significant decrease in MDA levels and an increase in SOD activity. This response implies that renalase anti-oxidative traits may be endorsed via the exogenous suppression of ROS formation and throughout the endogenous promotion of anti-oxidant competence, which warrants a degree of protection in cardiac pathology in relation to this phenomenon. Similarly, in experimentally induced chronic kidney failure [114], the administration of renalase ameliorates kidney and cardiac phenotypes observed as significantly reduced proteinuria, glomerular hypertrophy, renal interstitial fibrosis, hypertension, cardiomyocytes hypertrophy, cardiac interstitial fibrosis, and the prevention of cardiac remodeling through the inhibition of pro-fibrotic gene expression and ERK1/2 phosphorylation. Moreover, renalase supplementation significantly downregulates NADPH oxidase components, including gp91^phox^, p47^phox^, and p67^phox^. For the interpretation of these results in a more expedient manner, it must be noted that seven NADPH oxidase (NOXs) family enzymes (NOX1-5 and DUOX1-2) exist, which function as multipass transmembrane proteins that enzymatically generate superoxide via the transfer of electrons to molecular oxygen [135]. The NADPH oxidases NOX1 and NOX2, known as gp91^phox^ (as aforementioned), require the translocation of additional cytosolic subunits to the oxidase complex for activation. For instance, NOX1 requires association with the p47^phox^ or NOXO organizer subunit, the p67^phox^ or NOXA1 activator 130subunit, and the active (GTP-bound) Rho family small GTPase and Ras-related C3 botulinum toxin substrate 1 (Rac1), whereas NOX2 necessitates the interaction of p47^phox^, p67^phox^, p40^phox^, and active Rac1 or Rac2 with the NOX2: p22^phox^ complex at the membrane to evoke superoxide production [135,136]. In the setting of a failing heart, cardiomyocytes and several other cell types involved in the deterioration of cardiac function express the NOXs enzymes, which cross-link with mtROS to maintain cardiomyocyte growth, chronic inflammation, OS, microvascular dysfunction, and fibrosis, all of which are related to CD deterioration [136]. For instance, angiotensin II infusion-upregulated NOXs and mitochondrial oxidative damage, followed by significant cardiac hypertrophy, fibrosis, and diastolic dysfunction, may be improved through the administration of mitochondrial-targeted anti-oxidant agents [137]. Accordingly, it is acknowledged that targeting mitochondrial NADPH oxidases, which silences ROS generation, may favor some therapeutic welfare in heart failure patients [136]. Similarly, the inhibition of gp91^phox^ (which resembles the effects observed with renalase) mitigates OS and bolsters substantial myocardial protection to prevent CD progression [137]. If the proposition that NOXs represent another source of ROS in cardiomyocytes, inducing oxidative stress and redox signaling [135], is accepted, then it may be plausible that renalase, by downregulating the expression of gp91^phox^, p47^phox^, and p67^phox^ mediates the level of mitochondria-induced oxidative damage of cardiomyocytes. As a final note concerning the antioxidative potential of renalase, it must be highlighted that the ischemic preconditioning protection is likely mediated by renalase via the activation of TNFα/NF-κB signalization, as evidenced by the decreased MDA and increased SOD levels, diminished H_2_O_2_-induced apoptosis, and CD68 macrophage infiltration [115]. Likewise, there is evidence that renalase is upregulated via activation of the TNF-α/NF-κB pathway in skeletal muscles as a response to exercise-induced OS [36,138]. The most recent results, which can also be argued within the context of the mitochondrial protection of renalase, indicate that aerobic exercise increases the medullar expression of renalase in hypertensive rats [37]. Regular exercise is evidenced to be anti-apoptotic in nature via a mitochondrial-mediated apoptotic signaling network through the expert regulation of diverse apoptosis-regulating proteins. Exercise significantly impedes hypertension-induced cardiac Fas-dependent apoptotic pathways. Mitochondria-dependent apoptotic pathways thereby suppress mitochondria-mediated caspase-dependent apoptotic pathways and enhance cardiac pro-survival pathways [139,140,141]. It is conceivable that these effects are partly accomplished by renalase, given that aerobic exercise significantly increases renalase expression, followed by increased SOD and glutathione peroxidase activity, decreased levels of MDA, a decreased expression of NADPH oxidases 2/4 (NOXs2/4), and silenced apoptosis.

The aforementioned anti-oxidative nature of renalase, which is based on its traits evidenced in kidneys, may still prove applicable in CD, considering that the kidney also requires substantial mitochondrial content for energy production [86,142,143,144,145,146,147,148]. Nevertheless, CD progression is driven by some shared mechanisms with kidney disease: inflammation, OS, apoptosis, and mitochondrial dysfunction and interacts with kidney disease via various pathophysiological pathways. Accordingly, renal mtROS are known as one of the key drivers of impaired mitochondrial homeostasis and mitochondrial dysfunction, followed by decreased bioenergetics, weakened anti-oxidant defense, impaired protein quality control, diminished mtDNA repair function, defective mitophagy, and dynamic disturbances that have been acknowledged as pivotal mechanisms in diverse acute and chronic kidney pathologies, all of which lead to chronic kidney disease. The pathophysiology of mitochondrial dysfunction in kidney disease corresponds to that evidenced in the failing hearts scenario; therefore, it may be postulated that renalase exerts similar anti-oxidative effects in the heart as it does in kidneys. As a final note, it must be underscored that even if evidence is found that OS is a relevant pathophysiological feature during CD progression, clinical research of anti-oxidative stress therapies widely discloses unsatisfactory results [1,5,121]. It may be that the exclusive OS inhibition, by supplementation of exogenous anti-oxidants, may not be a sufficient approach for improved cardiac outcomes. Accordingly, there is a scientific rationale that renalase, owing to its multi-faceted traits, initially inhibits the production of the ROS and decreases MDA levels, followed by strengthening of endogenous anti-oxidant system activity (SOD, catalase, and GPH) and supplementing of cellular anti-oxidants (NAD^+^). If renalase concerts its actions to simultaneously bolster its endogenous anti-oxidant capacity in line with its capability to impede OS production, this should inspire further renalase scrutiny within the realm of cardiac disease.

## 6. The Intertwist of Renalase and Inflammation in Regard to Mitochondrial Dysfunction in Cardiac Disease

With the respect to the above, ample evidence exists that dysregulated mitochondria trigger and sustain an innate inflammatory response in failing hearts, thus significantly contributing to CD progression [5,8,10,13,149,150,151,152,153,154,155,156,157,158]. Sterile inflammation in the heart represents a response by which the cardiac tissue counteracts the acute injury to obtain myocardial adaptive remodeling [10,13,153]. However, when inflammation continues beyond the repair of the initial injury, it continuously alters the cardiac microenvironment (chronic low-grade inflammatory state) and becomes maladaptive. By working in conjunction with other detrimental pathways, it provides a setting for CD progression [151]. Upon myocardial injury, the majority of cardiac cells become quickly escalated and conjoined to set up a proinflammatory framework, serving as the origin of pro-inflammatory molecules (TNF-α, IL-6, IL-1). This initiates the accumulation of immune cells and their polarization towards pro-inflammatory subtypes, resulting in sustained innate inflammation [153]. If a cardiac injury induces mitochondrial dysfunction or damage, resulting in increased mtROS liberation, the immune response will be exaggerated by inappropriate mtROS generation, which is ascribed to have a role in CD progression [153,154]. Evidence indicates a complex cross-talk between inflammatory response and mtROS, with inflammation being triggered by mtROS [149,150,151,152,153,154] and vice versa. The mitochondrial role during inflammation is not solely that of being the source of mtROS; they also function as danger-associated molecular pattern (DAMP)-originating organelles [8,18,152,153]. In various settings, such as the opening of the mPTP or the outer mitochondrial membrane permeabilization, DAMPs are released into the cytosol, initiating inflammatory pathways that culminate in the disruption of normal cellular function. Damaged mtDNA, ATP, mitochondrial transcription factor A (TFAM), natural formylated peptides (NFP), succinate, and cardiolipin [18,149,155,156,157,158] have been identified as mitochondria-related DAMP molecules released upon cell death and known to initiate inflammation by activating cell surface receptors (P2X7R or FPRs) or intracellular receptors (TLR9 or NLRP3) [158]. Once released from damaged mitochondria, the mtDNA triggers a pro-inflammatory cascade by initiating the TLR9-mediated signaling network in immune cells for inflammatory cytokine production and inducing the adherence of neutrophils and endothelial cells through the activation of their adherence molecules (CD18 and ICAM-1). Neutrophils, upon the onset of NLRP3 inflammasome activation, upregulate IL-1R in cardiac fibroblasts to induce inflammation, apoptosis, and fibrosis [159], while mtDNA activates the NLRP3 inflammasome-mediated caspase-1 activation, which results in increased IL-1β and IL-18 secretion [13,18,149,154]. For a better understanding of the proposed crosstalk between inflammation, the NLRP3 inflammasome, and mitochondrial dysfunction, it must be emphasized that the NLPR3 inflammasome, expressed in macrophages, monocytes, and neutrophils [18], relates to an intracellular multiprotein complex physiologically placed in the cytosol. Upon the stimulation with mtDAMPs [149,154], it recruits the apoptosis-associated speck-like protein and caspase-1, resulting in a release of an IL-1β and IL-18 [18,149,154]. The specific macrophage pro-inflammatory phenotype (M1) is particularly abundant in NLRP3 inflammasome and, if extensively accumulated in the heart, produces a substantial amount of IL-1β [159], thereby aggravating cardiac inflammation. Therefore, the most intertwined and contributing interrelation of dysfunctional mitochondria (via DAMPs and ROS generation) and innate inflammatory response (via neutrophil and macrophage accumulation and NLRP3 inflammasome activation) emerges to sustain cardiac inflammation, thus contributing to CD progression [149,154,159]. An additional pathway, also significantly modulating NLRP3 inflammasome activity, is the mitochondrial redox state and NAD^+^ bioavailability [160], as discussed. Evidence indicates the role of the NRPL3 inflammasome in CD pathophysiology [8,160,161,162,163], given that its genetic ablation promotes improved cardiovascular health [8,160], presumably owing to the increased autophagic response. Accordingly, upon the pharmacological blockage of NLRP3 and IL-1β, systolic dysfunction and cardiac contractility were shown to be significantly improved [161]. Similarly, tNLRP3 activation and IL-1β secretion result in the increased expression of TGF-β1 in the left ventricle, followed by extracellular matrix accumulation, collagen I deposition, and adverse cardiac remodeling, along with caspase-1-mediated cell death [8,162]. Likewise, the samples of injured human hearts indicate increased levels of NRPL inflammasome-containing immune cells that purportedly contribute to CD pathophysiology [163].

As a final note, it must be pointed out that cardiac macrophages likely represent the pivotal cells for maintaining mitochondrial homeostasis in the steady-state setting and following cardiac injury [8,159,164]. Briefly, macrophages may be broadly subdivided into two general phenotypes: the M1 (pro-inflammatory characteristics) and M2 (associated with tissue repair and wound healing), whereas a shift in their population towards the M1 subclass may be accountable for cardiac function deterioration [164]. The M1 subclass is inherently enriched with inflammatory genes involved in the NLPR3 inflammasome assembly and IL-1β production [159]; a progressive decline in the M2 subpopulation is evidenced in cardiac hypertrophy and the vessel rarefaction phenotype, thereby enhancing cardiac inflammation and fibrosis [165]. Increasing evidence supports the hypothesis that the primary responders in sterile cardiac inflammation are macrophages, whereas mitochondrial dysfunction and OS significantly emerge in the setting of M1 polarization, apparently leading to NLRP3 inflammasome (over)activation, IL-1β secretion, and CD aggravation [8,159,164,165,166]. Despite the extensive rationale that sustained low-grade inflammation represents the initial point of CD progression, clinical trials which target inflammatory pathways yield unsatisfactory results [1,13], thus implying a need for a more optimized anti-inflammatory approach. In reference to that, several experimental studies in different cell cultures and in vivo evidenced the significant anti-inflammatory behavior of renalase in terms of pro-inflammatory cytokine and chemokine suppression, neutrophil accumulation, macrophage polarization hindering, and inflammasome silencing [30,32,44,61,113,114,115,116,117,118].

The initial concept that renalase suppresses an innate inflammatory drive, thus indirectly rescuing mitochondrial phenotype, arises from the findings that renalase gene expression is regulated by a range of transcriptional factors (NF-kB, TNF-α, and HIF-1α) [44,48,49,50,51,52], all correlating to systemic inflammation. This evidence indicates that renalase levels are likely to rise to counteract a proinflammatory cascade, given that the upregulation of renalase may be silenced by pretreatment with Humira (TNF-α antagonist) or pyrrolidine dithiocarbamate (specific NF-κB antagonist), thus leaving the tissue partially unprotected [115]. Accordingly, renalase, in vivo and in vitro, promotes cell survival by means of significantly reducing the influx of proinflammatory neutrophils into the tissue and abolishing total macrophage accumulation (CD68), including M1-like (CD86) and M2-like macrophages (CD163) [114]. Beyond solely reducing macrophage tissue accumulation, renalase regulates the M1/M2 shifting during disease progression by impeding M1-like macrophage polarization and promoting anti-inflammatory phenotype (M2) and suppressing proinflammatory gene expression: TNF-α, MCP-1, and the macrophage inflammatory protein (MIP)-2 [61,115,118]. It has been shown that cytoprotection is obtained through the engagement of renalase in various forms of downstream signaling, predominantly the Smad-independent network, such as PI3K/Akt and MAPK. The recognition that renalase, upon binding to its receptor, elicits downstream signals to modulate renalase-dependent MAPK pathways [48,61,112,116,119,120,134], provides plausibility for the proposition that the renalase/mitochondria axis may be upregulated to fine-tune the effects of MAPK activation on mitochondria [18,167]. For a deeper understanding of this hypothesis, MAPK signals (ERK1/2, p38, JNK) are intertwined to regulate cardiac development, metabolism, performance, and pathogenesis and to obtain a response to diverse cardiac stressors [167]. The MAPK network, through interactions with mitochondria (MAPK-mitochondria pathways), significantly impacts cardiomyocyte signaling in the setting of diverse cardiac pathologies, including myocardial infarction, ischemia/reperfusion injury, hypertrophy, and heart failure. Mitochondrial dysfunction per se also results in MAPK activation, implying a significant bidirectional relationship. Moreover, it is widely accepted that the mitochondria-mediated activation of MAPK associates with the mild stress-induced opening of mitoK_ATP_ channels in response to various cellular stresses, leading to a brief mtROS burst for activation of survival signaling molecules (cytoplasmic MAPK) and aiming to foster cardio-protection. In some cases, MAPK induce cardiomyocyte death, leading to overall cardiac dysfunction and disease progression. Sufficient evidence exists that the proper modulation of MAPK–mitochondria cross-talk may be a feasible approach in the prevention and treatment of CD [167]. It may be possible that this interaction is additionally modulated through renalase-dependent MAPK signaling to exert a context-dependent (state of the target cell or time-dependent) pleiotropic renalase response [131]. Additional findings relevant to this review indicate that renalase supplementation prevents deterioration of cardiac function and structure in terms of a lower degree of cardiomyocyte hypertrophy, ameliorated cardiac interstitial fibrosis (cardiac upregulation of matrix metalloproteinase (MMP)-1, down-regulation of tissue inhibitor of metalloproteinase (TIMP)-1 and TGF-β1, and silencing of ERK1/2 phosphorylation), decreased left ventricular (LV) end-diastolic posterior wall thickness, and LV end-diastolic pressure as well as a restored cardiac diastolic function [114]. Similarly, related research tested the hypothesis of whether renalase may elicit protective effects in the development of diabetic nephropathy [116]. It was confirmed that a lack of renalase results in increased arterial blood pressure, significant renal accumulation of inflammatory cells, renal mesangial expansion, and hypertrophy. There is also confirmation that renalase attenuates profibrotic gene expression and p21 expression through the inhibition of the ERK1/2 network; this is a commonly used protection strategy for renalase. The most recent results [117], which comprehensively correspond to the concept discussed in this review, evidenced that kidney-targeted renalase agonist administration (mesoscale nanoparticle (MNP)-encapsulated RP81) (RP81-MNP) results in the increased percentage of viable cells, followed by a reduction in the number of infiltrating neutrophils, CD4 T cells, dendritic cells, CD68 macrophages, M1 phenotype, myofibroblast activation, apoptotic cells, reduced ERK phosphorylation, activation of necroptosis molecules (RIPK1, RIPK3, and MLKL), and downregulated markers of kidney injury. In inflamed macrophages, the renalase agonist downregulates chemokines (Cxcl2, Ccl12, Ccr12, Ccl7), proinflammatory cytokines (IL-1β, TNF-α), complement/coagulation factors (C1qa, C1qb, C1qc), antigen process/presenting molecules, genes involved in toll-like receptor (TLR) signaling (Cd14, Spp1), and oxidative stress gene Gadd45. In terms of its systemic inflammatory response, the renalase agonist significantly reduces plasma levels of IL-1β, TNF-α, and IFN-γ. What is more captivating, the renalase agonist regulates genes encoding proteins involved in ferroptosis (Gpx4), autophagy (CD63), mitophagy (Ubb, Ubc, Cyba), stress (Ass1), and redox signaling (Prdx5); reduces TLR signaling and lysosome activation (CD63, Napsa); and modulates metabolic pathways (Comt, Fbp1, Miox). The most important revelation is that the renalase agonist upregulates mitochondrial genes involved in cellular vitality promotion (Cytb) and in the process of oxidative phosphorylation (Nd1, Nd3, Nd4L, Nd5). These mitochondrial genes encode subunits of NADH-ubiquinone oxidoreductase complex I, which represents the origin of ROS [117,168], thereby increasing the state of complex I reduction. This recognition illustrates that renalase, at least partially, regulates mitochondrial biogenesis and ameliorates mitochondrial dysfunction. Additionally, the transcription of the adhesion the G protein-coupled receptor E1 (Adgre1) gene, which encodes F4/80, a marker of mature macrophages, is significantly higher in the renalase knockout animals, thus implying enhanced macrophage tissue accumulation upon injury in the absence of the renalase gene [30]. Accordingly, the exposure of macrophages to renalase is followed by M2 shifting, while the same exposure of activated macrophages (M1) is found to silence the activation of the inflammasome, followed by suppressed IL-1β production [32,44]. Furthermore, renalase anti-inflammatory traits were evidenced in acute pancreatitis; the scarcity of renalase (knockout animals) results in significantly increased expression of pyknotic nuclei and apoptotic bodies in the pancreatic tissue and significant neutrophil and macrophage accumulation, followed by profound tissue deterioration [32]. Accordingly, the administration of recombinant renalase abrogates disease severity and inflammasome activation by triggering its receptor PMCA4b. Finally, it must be noted that the cells of innate immunity likely represent important targets for renalase [44], whereas the particular renalase-targeting receptor has been expressed on macrophages [169]. A schematic view of the potential anti-inflammatory effects of renalase against impaired mitochondrial homeostasis and cardiac inflammation is presented in Figure 2.

All things considered, renalase, apart from functioning as a regulator of proinflammatory cell influx and their tissue accumulation during the initial phase of the injury, conducts cell polarization into the anti-inflammatory (reparative) phenotype and hinders inflammasome activation and IL-1β secretion. These traits of renalase are likely assembled to supervise, fine-tune, or suppress the exaggerated and sustained innate immune response, thus resulting in nuanced and appropriate anti-inflammatory tissue protection. The innate immunity network is also highly subjected to mitochondrial regulation, with its cascade signaling which is dependent on mitochondria or at least requires some mitochondrial components [18]. As appealing as it is, this theory of a dialogue between renalase and mitochondria, in terms of shared signaling network and regulatory mechanisms to shape and regulate the nature of the cellular response to injury, is for the first time argued in this review. Whether a cause-and-effect relationship among renalase, cardiomyocyte mitochondrial dysfunction, and dysregulated innate immunity exists in CD pathophysiology and its extent, are yet to be fully scrutinized.

## 7. Conclusions

The CD burden remains a gargantuan public health issue and a rising global concern. Despite the progress of scientific discovery and the upgrade of other therapeutic research fields, CD continues to affect this ever-increasing population with poor quality of life and a slightly improved survival rate. Although current therapies provide some symptomatic and clinical welfare for the patients, they do not equally target the whole pathophysiological plethora of CD. Therefore, a novel concept should be embraced: the introduction of cardiomyocytes and their intracellular compartments, such as mitochondria, as novel points of therapeutic intervention.

Mitochondria are highly affected by any cardiac injury or damage, eventually leading to cardiomyocyte death, myocardial injury, and a failing heart. Therefore, it may be foreseen that these organelles are vastly involved in CD pathogenesis and that their manipulation for therapeutic gain may be a tantalizing possibility. The development of mitochondria-targeting therapies for CD demands a complete comprehension of mitochondrial functional pluralism, which eagerly searches for a multi-faceted molecule capable of simultaneously combating the pathophysiologic heterogeneity of mitochondria-induced cardiomyocyte injury, such as oxidative damage, reduced ATP production, apoptosis, and sustained activation of the innate immunity. 

Within this context, a shred of preclinical evidence exists that a novel enzyme/hormone known as renalase and functioning at various points of cellular damage, may counteract some of the pernicious effects caused by mitochondrial dysfunction. Renalase preserves mitochondrial integrity and dynamics, thus providing substantial mitochondrial quality control in terms of suppressing mitochondrial ΔΨm collapse, ameliorating overall ATP content, rising mtDNA copy numbers, and inhibiting mitochondrial translocation of the fission-related protein Drp-1. Moreover, renalase functions as an NADH oxidase for an efficient NAD^+^ supply to upregulate sirtuin activity, acting as an oxidative stress silencer for a decrease in mtROS and NADPH oxidase components (gp91^phox^, p47^phox^, and p67^phox^) and restoration of cellular antioxidative capacity (glutathione upregulation and increased SOD activity). The renalase agonist upregulates mitochondrial genes highly involved in cellular vitality promotion (Cytb) and oxidative phosphorylation (Nd1, Nd3, Nd4L, Nd5), thereby increasing the state of mitochondrial complex I reduction and oxidative stress amelioration. It also prevents oxidative injury-related consequences (ROS-induced EMT), hinders apoptosis (suppression of cytochrome C liberation, caspase-3 cleavage, Bax expression, and Bcl-2 induction) and regulates the innate inflammatory response (thwarting immune cell tissue accumulation and their polarization; suppressing proinflammatory cytokines and chemokines, such as IL-1β, TNF-α, IL-6, MCP-1, and MIP-2, as well as inflammasome activation and IL-1β secretion).

The multi-faceted nature of renalase pinpoints this protein at the regulatory interface of energy metabolism (intracellular renalase) and cellular mechanisms (extracellular renalase), thus providing a plausible relationship with mitochondria and exerting some therapeutic potential to target mitochondrial abnormalities in failing hearts. However, before any substantial conclusions about renalase and its relationship with mitochondria are completed, abundant questions and challenges must be resolved.

## Figures and Tables

**Figure 1 cells-12-01607-f001:**
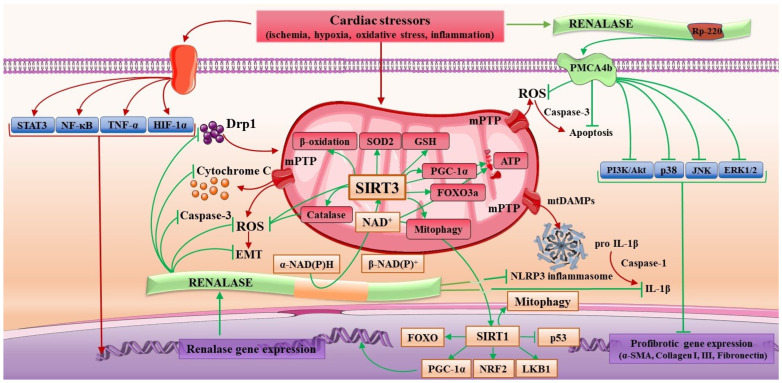
A schematic view of anticipated mechanisms of renalase in regard to mitochondrial protection upon various cardiomyocyte injuries. Renalase gene expression is regulated by a range of transcription factors, including STAT3, NF-κB, TNF-α, and HIF-1α, to acquire multi-faceted roles for renalase action. Evidence shows that renalase provides a state of antioxidation by suppressing cellular and mitochondrial ROS content and silencing its consequences (mtROS-induced epithelial to mesenchymal transition); upregulating GSH and SOD expression and abolishing MDA; downregulating NADPH oxidase components (gp91^phox^, p47^phox^, and p67^phox^) and a state of anti-apoptosis specifically by suppressing cytochrome C liberation, Bax expression, and cleaved caspase-3; and downregulating Bcl-2 expression. At the same time, renalase obtains the competence to enhance the phosphorylation of Drp-1 (the major fission protein) at serine 637 and to impede its translocation and its accumulation into mitochondria, thereby preventing mitochondrial fission. By that means, renalase efficiently prevents a mitochondrial ΔΨm collapse, ameliorates the overall ATP content, and increases mtDNA copy numbers to maintain mitochondrial integrity and dynamics. Moreover, renalase functions as an NADH oxidase (the amine oxidase domain is denoted in orange) by converting α-NAD(P)H to β-NAD(P)^+^, thereby ameliorating the cellular NAD^+^ levels and NAD^+^/NADH ratio and upregulating the activity of sirtuins (NAD^+^-dependent type III deacetylases). In that manner, renalase presumably provides an indirect role for promoting SIRT1-dependent mitochondrial biogenesis and mitophagy. For instance, SIRT1 promotes the nuclear accumulation of PGC-1α for the transcription of genes required for mitochondrial function and biogenesis. SIRT1 also activates (FOXO, NRF2, LKB1) or inhibits (p53) some signaling pathways to further inhibit oxidative stress, improve anti-oxidant protection, lessen inflammation, decrease inflammasome activity and IL-1β production, promote mitophagy, and suppress apoptosis for mitochondria quality control preservation and presumed cardioprotection against many cardiac stressors. Moreover, renalase promotes the activity of SIRT3, which, through deacetylation, regulates fatty acid oxidation; glucose and ketone body metabolism; amino acid catabolism; regulates oxidative phosphorylation and increases ATP production; governs oxidative stress by negative regulation of ROS production; preserves ΔΨm by increasing GSH and SOD levels; negatively regulates the mPTP opening; and stimulates mitophagy. SIRT3, for instance, similar to SIRT1, upregulates the expression of PGC1-α for mitochondrial function preservation and detoxification pathway activation; promotes mitochondrial FOXO3a for the transcription of anti-oxidative genes and genes necessary for mitochondrial biogenesis safety; and enhances the LKB1 pathway for ATP production. Collectively, by functioning as a mitochondrial homeostasis guardian, SIRT3 ameliorates the mitochondrial function of cardiomyocytes and increases energy production; hence, it is implicated in heart protection against myocardial ischemia, cardiac fibrosis, hypertrophy, atherosclerosis, dilated cardiomyopathy, and heart failure. However, extracellular renalase or its recombinant peptide, Rp-220 (denoted as red), upon targeting its receptor PMCA4b, elicits mediation of the vast downstream network, including PI3K/Akt, p38, JNK, ERK1/2, to obtain renalase-mediated cytoprotection, irrespective of the injury. The MAPKs signaling (ERK1/2, p38, JNK), specifically, mediate the activities of multiple cardiac stressors, being highly involved in cardiac development, metabolism, physiology, and pathogenesis. In terms of the implication of MAPK-mitochondria signaling in diverse cardiac pathologies (myocardial ischemia and infarction, hypertrophy, and heart failure), it may be that its modulation by renalase, at different levels, occurs in a context-dependent manner to provide some renalase-mediated mitochondria-induced cardiac protection. STAT3: signal transducer and activator of transcription 3; NF-κB: nuclear factor kappa B; HIF-1α: hypoxia-1α; ROS: reactive oxygen species; GSH: glutathione; SOD: superoxide dismutase; MDA: malondialdehyde; NADPH: nicotinamide adenine dinucleotide phosphate; Bax: Bcl-2-associated X protein; Bcl-2: B-cell lymphoma 2; Drp-1: dynamic-related protein-1; ΔΨm, changes in mitochondrial membrane potential; mPTP, mitochondrial permeability transition pore; NAD+: nicotinamide adenine dinucleotide; SIRT: sirtuin; NLRP3: NOD-like receptor family, pyrin domain containing 3; PGC-1α: peroxisome proliferator-activated receptor gamma coactivator-1α; FOXO: forkhead box O; NRF2: nuclear factor erythroid 2-related factor 2; LKB1: liver kinase B1; PMCA4b: plasma membrane calcium-ATPase 4b; ERK1/2, extracellular regulated protein kinases 1/2; PI3K/Akt, phosphatidylinositol 3-kinase/protein kinase B; JNK, c-Jun N-terminal kinases; MAPKs: mitogen-activated protein kinases.

**Figure 2 cells-12-01607-f002:**
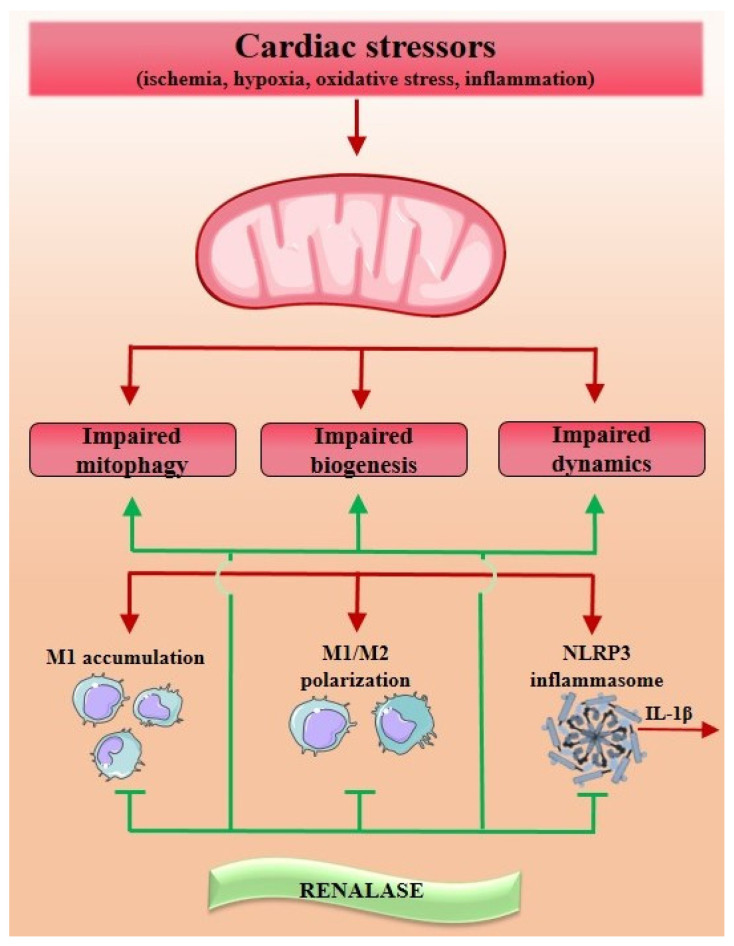
A schematic view of the potential anti-inflammatory effects of renalase against impaired mitochondrial homeostasis and cardiac inflammation upon cardiac injury. Diverse cardiac stressors (ischemia, hypoxia, and oxidative stress) significantly disrupt mitochondrial function, resulting in impaired mitochondrial homeostasis, which is characterized by suppressed mitophagy, reduced mitochondrial biogenesis, and dysregulated mitochondrial dynamics (fission and fusion). Such a setting triggers an innate inflammatory response as evidenced by myocardial accumulation of pro-inflammatory macrophages (M1), enhanced M1/M2 polarization, and assembly of the NLRP3 inflammasome, thereby aggravating cardiac inflammation and disease progression. Renalase regulates mitochondrial biogenesis, preserves mitochondrial integrity and dynamics, and, by upregulating the activity of sirtuins (Figure 1), provides an indirect role in sirtuin-dependent mitophagy and finally recovers mitochondrial quality control mechanisms. Renalase thwarts pro-inflammatory macrophage (M1) tissue accumulation and their polarization (M1/M2) and hinders inflammasome activation and IL-1β secretion, thereby mediating anti-inflammatory tissue protection. The multi-faceted nature of renalase suggests a proof of principle for the therapeutic potential of targeting mitochondrial abnormalities in failing hearts. NLRP3: NOD-like receptor family, pyrin domain containing 3.

**Table 1 cells-12-01607-t001:** The traits of renalase that may be productively endorsed for several mitochondria-mediating mechanisms in the pathophysiology of cardiac disease.

Research Model	Mainstay Findings	Therapeutic Gain	Ref
Acute low- and high-intensity treadmill exercise	Renalase expression is upregulated upon NF-κB or HIF-1α, depending on the type of the muscle fibers and the exercise intensity (acute low- and high-intensity training) and represents a response to exercise-induced oxidative stress.	Antioxidative skeletal muscles protection	[36]
Aerobic training in spontaneously hypertensive rats	Increased renalase in the renal medulla ameliorates the degree of oxidative stress (decreased NOX2 and NOX4 expression) and apoptosis (decreased Bax and cleaved-caspase-3 expression; increased Bcl-2 expression) upon aerobic exercise and improves hypertensive kidney injury.	Amelioration of hypertensive kidney injury	[37]
Ischemic myocardial damage	Renalase deficiency in knockout mice results in increased plasma catecholamine levels, hypertension, mild ventricular hypertrophy, and an increased degree of myocardial necrosis, while recombinant renalase supplementation significantly ameliorates cardiac injury. These cardio-protective traits are likely obtained through the ability of renalase to oxidize NADH, thereby efficiently recovering cellular NAD^+^ content and the NAD^+^/NADH ratio.	Mitigation of cardiac injury in renalase-deficient conditions	[59]
Cisplatin- and hydrogen peroxide -induced acute kidney injury	Renalase deficiency leads to a significant renal macrophage infiltration, acute tubular necrosis, and apoptosis (increased caspase-3 activation and decreased Bcl-2 expression), while renalase treatment reduces tissue phenotype via PI3K/Akt and MAPKs activation and JNK downregulation.	AKI protection in patients receiving cisplatin therapy	[61]
Cisplatin-induced acute kidney injury	Renalase administration upon acute injury regulates ROS generation and oxidative stress levels, increases phosphorylation of Drp1 at serine 637, decreases Drp1 translocation to mitochondria, and enhances SIRT3 expression, thereby ameliorating mitochondrial morphology and dynamics in a SIRT3-dependent manner.	Reno-protection in patients receiving cisplatin therapy	[111]
Unilateral ureteral obstruction	Renalase administration suppresses the levels of MDA and ROS, reinforces SOD, and impedes oxidative stress-mediated EMT by effectively decreasing α-SMA expression, fibronectin, and collagens, while recovering the expression of E-cadherin and renal interstitial fibrosis.	Anti-fibrotic effects and mitigation of chronic kidney disease	[112]
Contrast-induced nephropathy	Renalase pretreatment suppresses the inflammatory response by decreasing TNF-α and MCP-1 levels as well as macrophage tissue accumulation; ameliorates the level of histological injury; decreases MDA; improves SOD levels; and abolishes apoptosis.	Reno-protection in patients receiving contrast	[113]
Subtotal nephrectomy	The systemic delivery of renalase reduces renal infiltration of total macrophages (CD68), specifically M1-like (CD86) and M2-like macrophages (CD163), and silences M1/M2 polarization, pro-inflammatory cytokines (TNF-α, MCP-1, and IL-6), and NADPH oxidase component expression. Renalase administration rescues overall cardio–renal structure and function by preventing hypertrophy and fibrosis via a decrease in the expression of collagen I and III, TIMP-1, and TGF-β1 and an increase in the expression of MMP-1 through the inhibition of ERK1/2 signalization and pro-fibrotic gene expression.	Cardiovascular and renal protection in patients with CKD	[114]
Contrast-induced nephropathy	Limb ischemic preconditioning-induced reno-protection depends on renalase upregulation via the TNF-α/NF-κB signaling and results in significantly reduced macrophage accumulation, improved renal function, tubular damage, and oxidative stress mitigation.	Reno-protection in patients receiving contrast	[115]
Diabetic nephropathy	Renalase downregulation leads to significant renal inflammation, mesangial hypertrophy, kidney injury, hypertension, and albuminuria, whereas renalase administration mitigates profibrotic gene expression and p21 expression via the impediment of the ERK1/2 pathway.	Mitigation of diabetic nephropathy progression	[116]
Cisplatin-induced chronic kidney disease	The therapy with kidney-targeted renalase agonist (RP81-MNP) weakens the pro-inflammatory state of chronic kidney disease by hindering the renal accumulation of neutrophils, CD4 T cells, dendritic cells, M1 macrophages (CD68), and myofibroblasts. In inflamed macrophages (M1), RP81-MNP downregulates chemokines (Cxcl2, Ccl12, Ccr12, Ccl7), proinflammatory cytokines (IL-1β, TNF-α), complement/coagulation factors (C1qa, C1qb, C1qc), antigen process/presenting molecules (H2-Aa, K2-k1, Cd74), genes involved in TLR signaling (Cd14, Spp1), and oxidative stress gene Gadd45. Moreover, renalase agonist administration results in decreased proinflammatory cytokine plasma concentration (IFN-Υ, IL-1β, IL-6, and TNF-α), reduced number of apoptotic cells, inhibited renal necrosis, and preserved epithelial components of the nephron and the vasculature.	Reno-protection in patients receiving cisplatin therapy	[117]
Ischemic acute kidney injury	Renalase treatment, before the ischemic injury, lessens neutrophil and macrophage infiltration and modulates renal tubular necrosis and apoptosis, while the depletion of renalase increases the expression of the kidney’s proinflammatory genes (TNF-α, MCP-1, and MIP-2).	Biomarker, prevention, and therapy for ischemic acute kidney injury	[118]
Unilateralureteral obstruction	Renalase ameliorates renal interstitial fibrosis, evidenced by the maintenance of E-cadherin expression and α-SMA as well as fibronectin and collagen-I downregulation, by inhibiting the activation of the ERK1/2 signaling pathway.	Antifibrotic effects for slowing the progression of CKD	[119]
Transverse aortic constriction-induced heart failure	Renalase alleviates pressure overload-induced heart failure through p38 and ERK1/2 signaling	Biomarker of cardiac hypertrophy and therapy for heart failure	[120]

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
