# Peer review of "The Multi-Faceted Nature of Renalase for Mitochondrial Dysfunction Improvement in Cardiac Disease"

_cells, 2023, doi:10.3390/cells12121607_

Round 1
Reviewer 1 Report
This is an extensive, and useful, overview of the mechanism of action of renalase and its relevant derived peptides that the authors suggest may prove benefitial in the treatment of cardiac disease. The manuscript could benefit from extensive editing to shorten the review, and to point out what are merely speculations about how renalase works to improve outcome in other models of injury versus proven mehansims of action, especially those that support the hypothesis that renalase may participate in regenerative processes directly.
Author Response
Many thanks for your time and expertise.

Reviewer 2 Report
Comments
The submission (cells-2255251) is likely an unsolicited review article. The article describes mechanisms underlying the role of renalase in improving mitochondrial function or alleviating mitochondrial dysfunction during the progress of cardiac diseases. The topic of current review article is significant, however, the manuscript is not in good shape, which precludes us from considering it for publication in Cells. This reviewer has identified several flaws undermining the strength of the manuscript, which limits its merits and benefits to the scientific community.
1. The manuscript is too lengthy and inappropriately extended, which might be burdensome to readers. The way of writing should be more concise and focus on how the function of renalase regulates cardiac function as well as the disease mechanism of myocardial ischemia, myocardial ischemia and reperfusion injury, and heart failure, conveying key concepts to general readers. Authors should avoid too many redundant details out of the scope of the main theme, such as fatty liver disease, steatohepatitis, hepatic I/R, acute kidney injury, GI injury etc (Tables 1 & 2) which are not within the scope of cardiac disease. The length of manuscript can be reduced by 30-40%.
2. The manuscript did present an overall thematic picture or a main thematic scheme (Figure 2) which is the strength of the manuscript. However, readers may have a difficult time ascertaining the focus and how the signal pathways centered at renalase regulates the mechanism of cardiac diseases. This needs to be improved and further clarified in the current review article.
3. The field of cardiac disease is relatively sizeable and the disease mechanisms are quite diversified. As the underlying mechanisms of current review emphasize renalase and cardiac mitochondrial dysfunction. It is better to be as focused and specific as possible in the topic to develop a significant review article. The scope and content of the current submission covers wide-ranging aspects, including facets outside the scope of mitochondrial biology, which is sporadic and could be potentially confusing to reader.
Author Response

(The authors gave the same response as above.)
